# Zika Virus infection and Guillain-Barré syndrome in Northeastern Mexico: A case-control study

**Fernando Gongora-Rivera**[1], **Israel Grijalva**[2], **Adrian Infante-Valenzuela**[1], **Carlos Cámara-Lemarroy**[1], **Elvira Garza-González**[3], **Martin Paredes-Cruz**[2], **Concepción Grajales-Muñiz**[4], **José Guerrero-Cantera**[2], **Ignacio Vargas-Ramos**[5], **Jesus Soares**[6], **Joseph Y. Abrams**[6], **Ashley R. Styczynski**[7], **Adrián Camacho-Ortiz**[8], **Margarita E. Villarino**[9], **Ermias D. Belay**[6], **Lawrence B. Schonberger**[6], **James J. Sejvar**[6]*, **GBS HU-IMSS Working Group**[¶]

1 Department of Neurology, University Hospital José Eleuterio González, Universidad Autonoma de Nuevo León, Monterrey, Nuevo León, Mexico, 2 Medical Research Unit for Neurological Diseases, UMAE Hospital de Especialidades, Centro Médico Nacional Siglo XXI, Instituto Mexicano del Seguro Social, Mexico City, Mexico, 3 Department of Gastroenterology, University Hospital José Eleuterio González, Universidad Autonoma de Nuevo León, Monterrey, Nuevo León, Mexico, 4 Epidemiological Surveillance Division for Transmissible Diseases, Epidemiological Surveillance Coordination, Instituto Mexicano del Seguro Social, Mexico City, Mexico, 5 Department of Neurology, UMAE Hospital de Especialidades No. 25, Instituto Mexicano del Seguro Social, Monterrey, Nuevo León, Mexico, 6 Division of High-Consequence Pathogens and Pathology—Centers for Disease Control and Prevention, Prion and Public Health Office, Atlanta, Georgia, United States of America, 7 Department of Infectious Disease, Stanford University, Palo Alto, California, United States of America, 8 Department of Infectious Disease, University Hospital José Eleuterio González, Universidad Autonoma de Nuevo León, Monterrey, Nuevo León, Mexico, 9 Division of Global Migration and Quarantine, CDC Mexico Country Office, Centers for Disease Control and Prevention, Atlanta, Georgia, United States of America

¶ Membership of the GBS HU-IMSS Working Group is provided in the Acknowledgments.
* zea3@cdc.gov

**Data Availability Statement:** The data underlying the results presented in the study are available from Facultad de Medicina y Hosp Universitario de

## Abstract

### Background

Beginning August 2017, we conducted a prospective case-control investigation in Monterrey, Mexico to assess the association between Zika virus (ZIKV) and Guillain-Barré syndrome (GBS).

### Methods

For each of 50 GBS case-patients, we enrolled 2–3 afebrile controls (141 controls in total) matched by sex, age group, and presentation to same hospital within 7 days.

### Results

PCR results for ZIKV in blood and/or urine were available on all subjects; serum ZIKV IgM antibody for 52% of case-patients and 80% of controls. Subjects were asked about antecedent illness in the two months prior to neurological onset (for case-patients) or interview (for controls). Laboratory evidence of ZIKV infection alone (PCR+ or IgM+) was not significantly different between case-patients and controls (OR: 1.26, 95% CI: 0.45–3.54) but antecedent symptomatic ZIKV infection [a typical ZIKV symptom (rash, joint pain, or conjunctivitis) plus

la UANL. Contact information: Wendy Gonzalez, Research Coordinator - Department of Neurology, University Hospital José Eleuterio Gonzalez, Universidad Autonoma de Nuevo Leon, Monterrey, Nuevo Leon, Mexico Email: wendygzz_inv. clinica@hotmail.com +818 33373 13 +811 3578 834

**Funding:** This study was supported by the United States Centers for Disease Control and Prevention. The funders had no role in study design, data collection and analysis, decision to publish, or preparation of the manuscript.

**Competing interests:** The authors have declared that no competing interests exist.

laboratory evidence of ZIKV infection] was higher among case-patients (OR: 12.45, 95% CI: 1.45–106.64). GBS case-patients with laboratory evidence of ZIKV infection were significantly more likely to have had typical ZIKV symptoms than controls with laboratory evidence of ZIKV infection (OR: 17.5, 95% CI: 3.2–96.6). This association remained significant even when only GBS case-patients who were afebrile for 5 days before onset were included in the analysis, (OR 9.57 (95% CI: 1.07 to 85.35).

## Conclusions

During ZIKV epidemics, this study indicates that increases in GBS will occur primarily among those with antecedent symptomatic ZIKV.

## Introduction

Guillain-Barré syndrome (GBS) is an acute inflammatory immune-mediated polyradiculo-neuropathy presenting classically with ascending progressive weakness, sensory changes, and hyporeflexia [1]. It is the most common cause of acute flaccid paralysis worldwide [1, 2]. GBS is usually precipitated by a preceding infection or other antigenic stimuli [3]. One of the pathophysiological mechanisms linked to GBS is molecular mimicry, as some pathogens may have antigens similar to peripheral nerve myelin and/or axonal epitopes [1, 2, 4]. The most commonly identified triggering agents are *Campylobacter jejuni*, cytomegalovirus, Epstein-Barr virus, and *Mycoplasma pneumoniae* [1, 2, 4].

Arthropod-borne viruses (arboviruses) such as Zika virus (ZIKV), dengue virus (DENV) and chikungunya virus (CHIKV) have become an increasingly important global health threat. DENV and ZIKV are arboviruses belonging to the *Flavivirus* genus of the *Flaviviridae* family, and are both transmitted by the *Aedes* species mosquito vectors [5, 6]. CHIKV is an arbovirus belonging to the alphavirus genus of the *Togaviridae* family [6]. Some arboviruses have been temporally associated with the occurrence of GBS [5].

A case-control study carried out during the French Polynesia ZIKV outbreak in 2013–2014 was the first to demonstrate an association between ZIKV and a large increase in the incidence of GBS [7]. Subsequently, ZIKV was introduced to South and Central America where excess reports of GBS were also reported in ZIKV-affected areas, and where DENV and CHIKV were already endemic [8, 9]. In Mexico, the first cases of ZIKV were reported in late 2015 [10, 11]. In this study, we assessed whether post-outbreak endemic circulation of ZIKV in Northeastern Mexico was associated with the development of GBS.

## Methods

### Ethics statement

The Institutional Review Board at the Universidad Autonoma de Nuevo Leon reviewed this protocol and approved it as research. The U.S. Centers for Disease Control and Prevention (CDC) relied on the IRB determination of UANL. All subjects provided written informed consent prior to investigation participation. The investigation was financially supported by CDC.

### Study design and participants

We conducted a case-control study in the northeast of Mexico, including Coahuila, Nuevo León and Tamaulipas States. The study period was from August 1, 2017 to June 30, 2018. The

study protocol was approved by the institutional review boards of Universidad de Nuevo León-Hospital Universitario (HU), the Instituto Mexicano del Seguro Social (IMSS), and the Centers for Disease Control and Prevention (CDC) before recruitment of GBS patients and controls. Informed consent was obtained from all GBS patients and control subjects before inclusion in the study. Patients were enrolled from three referral hospitals (emergency department visits or inpatient wards) of Monterrey City metropolitan area.

We identified suspected GBS case-patients based on onset of compatible neurologic symptoms (e.g., flaccid limb weakness, areflexia, cranial nerve palsies) reported by physicians and hospitals to a committee of investigators during the study period. To verify a GBS diagnosis, we performed medical record reviews to ascertain characteristics of the clinical illness and diagnostic testing, including cerebrospinal fluid, neuroimaging, and electro diagnostic test results, if available. Suspected GBS case-patients were classified according to diagnostic certainty of the Brighton Collaboration criteria case definitions for GBS [12]. Case-patients meeting levels 1–3 of diagnostic certainty were classified as confirmed GBS and eligible for enrollment in the investigation.

For each GBS case-patient, we enrolled three controls from the same hospitals seen in the emergency department or inpatient service within seven days of the GBS case with a non-febrile illness (no report or documentation of fever 48 hours before enrolment) that were matched to case-patients by sex and age ±10 years.

We interviewed all available case-patients and controls to obtain information about demographics, risk factors (age, male sex), and exposures in the two months prior to interview, for controls or to onset of neurological symptoms for the GBS case-patients. Functional outcomes in patients with GBS were assessed based on residual motor deficits using the Hughes GBS Disability Scale. Following the interviews, serum and urine samples were collected from case-patients and controls, to determine exposure to ZIKV, DENV and CHIKV.

## Laboratory analysis

**Viral RNA detection.** Viral RNA was extracted from the serum and urine samples by using the QIAamp Viral RNA Mini Kit (QIAGEN, Hilden, Germany). The Superscript III Platinum OneStep Real Time RT-PCR system (Invitrogen, Carlsbad, CA, USA) was used for RNA amplification to analyze gene expression. PCRs specific for DENV, CHIKV, and ZIKV were performed using the CDC Trioplex Real-time RT-PCR Assay (Trioplex Real-time RT-PCR Assay, method available at http://www.fda.gov/downloads/MedicalDevices/Safety/EmergencySituations/UCM491592.pdf).

**IgM antibodies against ZIKV.** IgM antibodies against ZIKV were determined by the use of the InBios ZIKV via an IgM antibody capture enzyme-linked immunosorbent assay (InBios MAC-ELISA; InBios International, Inc., Seattle, WA). InBios ELISA was performed and results interpreted as described by the manufacturer. An immune status ratio (ISR) was determined by dividing the OD of the patient sample with the ZIKV recombinant. ISR values of 1.7 were considered presumptive positive for IgM antibodies to ZIKV.

## Statistical analysis

To determine a possible association between GBS and a preceding ZIKV infection, we estimated that 49 case-patients and 147 controls would provide a power of 80% to detect a difference of 20% in ZIKV prevalence, with an alpha level of 5%. Descriptive statistics was used to summarize clinical and demographic data. We conducted analyses assessing potential associations between GBS and demographic characteristics, known GBS risk factors, antecedent illness in the two months prior to hospitalization, and molecular and serological evidence of

ZIKV infections. To assess possible differences between GBS case-patients and the matched controls, we calculated matched odds ratios and 95% confidence intervals by conditional maximum likelihood estimation, aside from comparisons for which these calculations did not converge, for which we calculated unconditional maximum likelihood estimates with confidence intervals produced using normal approximation. For comparisons between ZIKV positive and ZIKV negative GBS case-patients, we calculated unconditional maximum likelihood estimates with confidence intervals produced using normal approximation. For comparisons with zero values in any cells (such that odds ratio calculations were not calculable) we assessed differences using Fisher exact p-values (p ≤ 0.05 was considered statistically significant). We considered the presence of one or more of the following three antecedent symptoms–rash, joint pain and/or conjunctivitis–as having "typical" Zika symptoms.

IgM antibody testing was not available to be performed for all GBS case-patients and controls. Therefore, we used two different measures to assess ZIKV status by laboratory testing:

- Positive PCR assay (all patient in cohort included in analysis)

- Positive PCR assay or positive IgM assay (only patients with available IgM results used in analysis)

To assess antecedent symptomatic ZIKV infection, we used the following measures:

- Positive PCR assay and at least one of the typical Zika symptoms (all patients in cohort included in analysis)

- Positive PCR or IgM assay, and at least one of the typical Zika symptoms (only patients with available IgM results used in analysis)

All data were analyzed using R version 3.3.3 (The R Foundation for Statistical Computing, 2017).

## Results

During the study period, 50 GBS case-patients, and 141 (24 outpatient, 117 hospitalized) controls were enrolled. Demographics of the case-patients and controls as well as prevalence of virus infection is shown in Table 1.

Current infection by arboviruses (supported by the detection of ZIKV, DENV and/or CHIKV qRT-PCR) was not significantly different between groups (Table 1).

ZIKV was the most commonly detected infection, in 22% each of case-patients and controls (Table 1). IgM antibody testing for ZIKV was performed for 26 of 50 GBS case-patients (52%) and 113 of 141 controls (80%) (Table 2).

The median times between symptom onset and sample collection did not significantly differ by ZIKV status (Table 3).

In the two months before their admission, case-patients reported a variety of symptoms, including typical symptoms of ZIKV infection such as rash, joint pain and conjunctivitis (Table 4).

When comparing the rates of previous illness, case-patients reported typical ZIKV symptoms more frequently than controls (OR: 9.58, 95% CI: 3.16–29.09) (Table 4). Of GBS case-patients, 38.5% had evidence of ZIKV by PCR or IgM, compared to 30.1% of controls (OR: 1.26, 95% CI: 0.45–3.54). Case-patients were more likely than controls to have laboratory evidence of ZIKV infection in conjunction with a history of typical ZIKV symptoms (OR: 12.45, 95% CI: 1.45–106.64) ("symptomatic ZIKV"; Table 5).

Of the 16 GBS case-patients with an antecedent typical ZIKV symptom, six (38%) had a positive PCR test for ZIKV; none had a positive PCR test for DENV or CHIKV. For GBS case-

**Table 1. Demographics, geographic origin and virus infection.**

| | Case-patients, n = 50 | Controls, n = 141 |
|---|---|---|
| **Demographics** | | |
| Median age (range) | 40.5 (3–66) | 40 (2–70) |
| Male n (%) | 31 (62) | 90 (63.8) |
| **State of origin n (%)** | | |
| Nuevo León | 32 (64) | 99 (70.2) |
| Coahuila | 9 (18) | 23 (16.3) |
| Tamaulipas | 9 (18) | 18 (12.8) |
| San Luis Potosi | 0 (0) | 1 (.7) |
| **Virus infection (PCR+) n (%)** | | |
| ZIKV+ n (%) | 11 (22) | 31 (22) |
| DENV+ n (%) | 1 (2) | 2 (1) |
| CHIKV+ n (%) | 1 (2) | 7 (5) |
| **Antibody response (IgM+)** | | |
| ZIKV+ n (%) | 3 (12)* | 9 (8)[&] |

\* out of 26 case-patients

[&] out of 113 controls

patients, seven of 10 (70%) that had laboratory evidence for ZIKV infection by PCR or IgM also had typical ZIKV symptoms compared to two of 16 (13%) of those that tested negative for ZIKV (OR: 16.3, 95% CI: 2.2–121). In comparison, only four of 34 (12%) controls that had tested positive for ZIKV had typical ZIKV symptoms compared to five of 79 (6%) that tested negative for ZIKV (OR: 2.0, 95% CI 0.50–7.9). GBS case-patients with laboratory evidence of ZIKV infection were significantly more likely to have had typical ZIKV symptoms than controls with laboratory evidence of ZIKV infection (OR: 17.5, 95% CI: 3.2–96.6). This association remained statistically significant even in an analysis that included only case-patients with no febrile illnesses within 5 days prior to onset of GBS (OR 9.57 (95% CI: 1.07 to 85.35).

The majority of GBS case-patients-patients had paresis and areflexia, and 22% had facial diplegia (Table 6).

**Table 2. Number of GBS case-patients and matched controls by results of Zika PCR test, Zika IgM test, and presence of Zika symptoms.**

| | | GBS case-patients (n = 50) | | | |
|---|---|---|---|---|---|
| | | IgM assay performed (n = 26) | | IgM assay not performed | Total |
| | | IgM positive | IgM negative | | |
| PCR positive | ≥1 Zika symptoms | 0 | 4 | 2 | 11 |
| | No Zika symptoms | 0 | 3 | 2 | |
| PCR negative | ≥1 Zika symptoms | 3 | 2 | 5 | 39 |
| | No Zika symptoms | 0 | 14 | 15 | |
| | | Matched controls (n = 141) | | | |
| | | IgM assay performed (n = 113) | | IgM assay not performed | Total |
| | | IgM positive | IgM negative | | |
| PCR positive | ≥1 Zika symptoms | 0 | 3 | 0 | 31 |
| | No Zika symptoms | 3 | 22 | 3 | |
| PCR negative | ≥1 Zika symptoms | 1 | 5 | 0 | 110 |
| | No Zika symptoms | 5 | 74 | 25 | |

**Table 3. Time (days) from neurological symptom onset to sample collection from 50 case-patients by Zika status.**

| Category | | | N | Median | IQR | Mean | Min | Max |
|---|---|---|---|---|---|---|---|---|
| | | | | | **Days from neuro onset to sample collection** | | | |
| **All case-patients** | | | 50 | 11 | (7–17) | 13.46 | 2 | 52 |
| PCR only | Zika+ | | 11 | 15 | (9–23.5) | 19.27 | 5 | 52 |
| | Zika- | | 39 | 11 | (7–16.5) | 11.82 | 2 | 24 |
| PCR and rash, joint pain, or conjunctivitis | Zika+ | | 6 | 20.5 | (11.25–35) | 24.83 | 8 | 52 |
| | Zika- | | 44 | 11 | (7–17) | 11.91 | 2 | 24 |
| **All case patients who received an IgM test** | | | 26 | | | | | |
| PCR or IgM | Zika+ | | 10 | 17.5 | (8.5–20.5) | 19 | 2 | 52 |
| | Zika- | | 16 | 14 | (9–18) | 14.19 | 5 | 24 |
| PCR or IgM and rash, joint pain, or conjunctivitis | Zika+ | | 7 | 17 | (9–28) | 20.71 | 2 | 52 |
| | Zika- | | 19 | 15 | (9–20) | 14.32 | 5 | 24 |

Overall, there were few clinical differences between GBS patients with laboratory evidence of recent ZIKV infection and those without. Patients with both laboratory evidence of ZIKV infection and at least one antecedent typical ZIKV symptom ("symptomatic ZIKV") reported dyspnea more frequently (43% vs 5%, OR: 13.50, 95% CI: 1.10–165.97 (Table 6). Hughes score at nadir was not significantly different between ZIKV+ and ZIKV- GBS case-patients (Table 7).

**Table 4. Association between prior illness and GBS for 50 GBS case-patients and 141 matched controls–clinical symptoms reported prior to neurological onset/interview and virus infection.**

| Prior illness | Numbers (%) reported with antecedent symptoms | | Matched odds ratio (cMLE) | | |
|---|---|---|---|---|---|
| | Case-patients, n = 50 | Controls, n = 141 | Estimate | LL | UL |
| **Fever** | 16 (32) | 13 (9.2) | 5.93 | 2.27 | 15.50 |
| **Chills** | 2 (4) | 11 (7.8) | 0.50 | 0.10 | 2.37 |
| **Nausea** | 4 (8) | 17 (12.1) | 0.65 | 0.20 | 2.16 |
| **Diarrhea** | 22 (44) | 5 (3.5) | 12.17 | 4.60 | 32.19 |
| **Muscle Pain** | 8 (16) | 12 (8.5) | 2.84 | 0.87 | 9.25 |
| **Joint Pain** | 8 (16) | 8 (5.7) | 3.49 | 1.18 | 10.27 |
| **Skin Rash** | 6 (12) | 1 (.7) | 17.10 | 2.05 | 142.37 |
| **Conjunctivitis** | 6 (12) | 1 (.7) | 18.00 | 2.17 | 149.51 |
| **Headache** | 10 (20) | 13 (9.2) | 3.00 | 1.10 | 8.22 |
| **Retro Ocular Pain**[1] | 4 (8) | 0 (0.0) | - | - | - |
| **Nuchal Rigidity**[1] | 0 (0) | 1 (0.7) | - | - | - |
| **Confusion**[1] | 1 (2) | 0 (0.0) | - | - | - |
| **Abdominal Pain** | 8 (16) | 11 (7.8) | 2.16 | 0.85 | 5.52 |
| **Cough** | 9 (18) | 11 (7.8) | 3.08 | 1.11 | 8.53 |
| **Nasal Secretion** | 5 (10) | 5 (3.5) | 3.00 | 0.87 | 10.36 |
| **Odynophagia**[2] | 3 (6) | 1 (0.7) | 8.94 | 0.91 | 87.99 |
| **Periarticular Edema**[1] | 0 (0) | 2 (1.4) | - | - | - |
| **Lower Back Pain** | 1 (2) | 3 (2.1) | 1.00 | 0.08 | 11.93 |
| **Typical Zika symptoms**[3] | 16 (32) | 9 (6.4) | 9.58 | 3.16 | 29.09 |

[1]Undefined odds ratio/confidence limits.

[2]Odds ratio and confidence limits calculated by unconditional maximum likelihood estimation with normal approximation.

[3]Any of the following: rash, joint pain, conjunctivitis

**Table 5. Association between prior illness and GBS for 50 GBS case-patients and 141 matched controls–laboratory tests.**

| Prior illness | Number (%) | | Matched odds ratio (cMLE) | | |
|---|---|---|---|---|---|
| | Case-patients | Controls | Estimate | LL | UL |
| **All observations (50 case-patients, 141 controls)** | | | | | |
| Zika (PCR) | 11 (22) | 31 (22.0) | 1.03 | 0.44 | 2.39 |
| Zika (PCR) w/o typical symptoms[1] | 5 (10) | 28 (19.9) | 0.42 | 0.14 | 1.26 |
| Zika (PCR) with typical symptoms[2] | 6 (12) | 3 (2.1) | 14.26 | 1.68 | 120.98 |
| **Patients receiving IgM tests (26 case-patients, 113 controls)** | | | | | |
| Zika (PCR or IgM) | 10 (38.5) | 34 (30.1) | 1.26 | 0.45 | 3.54 |
| Zika (PCR or IgM) w/o typical symptoms[1] | 3 (11.5) | 30 (26.5) | 0.41 | 0.11 | 1.45 |
| Zika (PCR or IgM) with typical symptoms[2] | 7 (26.9) | 4 (3.5) | 12.45 | 1.45 | 106.64 |

[1] Laboratory evidence of Zika but none of the following: rash, joint pain, conjunctivitis.

[2] Laboratory evidence of Zika with any of the following: rash, joint pain, conjunctivitis.

In total, 48% had complete neurophysiological studies for analysis. These revealed a predominance of AMAN compared to demyelinating subtype (Table 7). Treatment was initiated with intravenous immunoglobulin in 68% and plasmapheresis in 20% (Table 7). Mechanical ventilation was required in 12% of patients, and significantly more ZIKV-positive case-patients (diagnosed by PCR and at least one antecedent typical ZIKV symptom) required mechanical ventilation than ZIKV negative case-patients (OR: 13.67, 95% CI: 1.88–99.35). There was one in-hospital death after a long stay in the intensive care unit.

**Table 6. Neurological signs and symptoms at onset or nadir of GBS case-patients: n (%).**

| Neurological signs and symptoms | All (n = 50) | Zika diagnosis by PCR or IgM (n = 26)* | | | Zika diagnosis by PCR or IgM and rash, joint pain, or conjunctivitis (n = 26)* | | |
|---|---|---|---|---|---|---|---|
| | | Zika+ (n = 10) | Zika- (n = 16) | Odds Ratio | Zika+ (n = 7) | Zika- (n = 19) | Odds Ratio |
| **Acute bilateral paresis** | | | | | | | |
| Upper extremities | 46 (92.0) | 8 (80.0) | 16 (100.0) | - | 6 (85.7) | 18 (94.7) | 0.33 (0.02–6.19) |
| Lower extremities | 46 (92.0) | 8 (80.0) | 15 (93.8) | 0.27 (0.02–3.41) | 6 (85.7) | 17 (89.5) | 0.71 (0.05–9.27) |
| **Areflexia** | | | | | | | |
| Upper extremities | 47 (94.0) | 9 (90.0) | 14 (87.5) | 1.29 (0.10–16.34) | 6 (85.7) | 17 (89.5) | 0.71 (0.05–9.27) |
| Lower extremities | 47 (94.0) | 9 (90.0) | 14 (87.5) | 1.29 (0.10–16.34) | 6 (85.7) | 17 (89.5) | 0.71 (0.05–9.27) |
| **Paresthesia/Sensory changes** | | | | | | | |
| Upper extremities | 19 (38.0) | 1 (10.0) | 8 (50.0) | 0.11 (0.01–1.09) | 1 (14.3) | 8 (42.1) | 0.23 (0.02–2.30) |
| Lower extremities | 21 (42.0) | 4 (40.0) | 7 (43.8) | 0.86 (0.17–4.27) | 3 (42.9) | 8 (42.1) | 1.03 (0.18–5.95) |
| **Dyspnea** | 11 (22.0) | 3 (30.0) | 1 (6.2) | 6.43 (0.56–73.35) | 3 (42.9) | 1 (5.3) | 13.50 (1.10–165.97) |
| **Facial diplegia** | 11 (22.0) | 3 (30.0) | 4 (25.0) | 1.29 (0.22–7.50) | 3 (42.9) | 4 (21.1) | 2.81 (0.44–18.06) |
| **Dysphagia** | 6 (12.0) | 2 (20.0) | 1 (6.2) | 3.75 (0.29–47.99) | 2 (28.6) | 1 (5.3) | 7.20 (0.54–96.64) |
| **Ophthalmoparesis** | 12 (24.0) | 2 (20.0) | 5 (31.2) | 0.55 (0.08–3.59) | 1 (14.3) | 6 (31.6) | 0.36 (0.04–3.70) |
| **Dysarthria** | 4 (8.0) | 2 (20.0) | 0 (0.0) | - | 2 (28.6) | 0 (0.0) | - |
| **Ataxia** | 4 (8.0) | 2 (20.0) | 1 (6.2) | 3.75 (0.29–47.99) | 1 (14.3) | 2 (10.5) | 1.42 (0.11–18.59) |
| **Dysautonomia** | 1 (2.0) | 0 (0.0) | 0 (0.0) | - | 0 (0.0) | 0 (0.0) | - |

* Of the 50 case-patients, 26 had IgM testing done, and Zika diagnosis was determined for these 26 case-patients by either: 1) positive PCR or positive IgM test, or 2) positive PCR or positive IgM test, and one of the following symptoms: rash, conjunctivitis, joint pain.

Table 7. Treatment and clinical results for GBS case-patients by Zika status: n (%).

| Treatment and clinical results | All (n = 50) | Zika diagnosis by PCR or IgM (n = 26)* | | | Zika diagnosis by PCR or IgM and rash, joint pain, or conjunctivitis (n = 26)* | | |
|---|---|---|---|---|---|---|---|
| | | Zika+ (n = 10) | Zika- (n = 16) | Odds Ratio | Zika+ (n = 7) | Zika- (n = 19) | Odds Ratio |
| Intravenous immunoglobulin | 34 (68.0) | 6 (60.0) | 10 (62.5) | 0.90 (0.18–4.55) | 3 (42.9) | 13 (68.4) | 0.35 (0.06–2.06) |
| Plasma exchange | 10 (20.0) | 2 (20.0) | 4 (25.0) | 0.75 (0.11–5.11) | 2 (28.6) | 4 (21.1) | 1.50 (0.21–10.82) |
| Mechanical ventilation | 6 (12.0) | 1 (10.0) | 0 (0.0) | - | 1 (14.3) | 0 (0.0) | - |
| Hughes score at nadir: mean (SD) | 3.5 (1.1) | 3.2 (1.2) | 3.4 (0.7) | - | 3.3 (1.4) | 3.4 (0.8) | - |
| Neurophysiological study | | | | | | | |
| AMAN[1] | 10 (20.0) | 1 (10.0) | 3 (18.8) | 0.48 (0.04–5.40) | 1 (14.3) | 3 (15.8) | 0.89 (0.08–10.30) |
| AIDP[2] | 6 (12.0) | 1 (10.0) | 2 (12.5) | 0.78 (0.06–9.88) | 1 (14.3) | 2 (10.5) | 1.42 (0.11–18.59) |
| AMSAN[3] | 4 (8.0) | 0 (0.0) | 2 (12.5) | - | 0 (0.0) | 2 (10.5) | - |
| Other | 4 (8.0) | 2 (20.0) | 2 (12.5) | 1.75 (0.21–14.93) | 1 (14.3) | 3 (15.8) | 0.89 (0.08–10.30) |

[1]AMAN = Acute motor axonal neuropathy.

[2]AIDP = Acute inflammatory demyelinating polyneuropathy.

[3]AMSAN = Acute motor and sensory axonal neuropathy.

* Of the 50 case-patients, 26 had IgM testing done, and Zika diagnosis was determined for these 26 case-patients by either: 1) positive PCR or positive IgM test, or 2) positive PCR or positive IgM test, and one of the following symptoms: rash, conjunctivitis, joint pain.

## Discussion

Our study suggests that symptomatic ZIKV infection (laboratory evidence of ZIKV infection plus one or more typical symptoms) but not asymptomatic ZIKV infection was associated with GBS compared to controls. When we combined laboratory evidence of ZIKV infection and the presence of typical symptoms of ZIKV, there appeared to be more symptomatic ZIKV case-patients in the GBS group than in the control group. Furthermore, this subgroup also showed some subtle differences in their clinical presentation. The only other study in which symptoms and laboratory evidence consistent with a ZIKV illness were combined in order to assess an association of increased ZIKV with increased cases of GBS was one conducted in Brazil after ZIKV was first introduced into that country. The previous study reported no significant association between recent *Flavivirus* infection (a positive or equivocal IgM test result for ZIKV or DENV) and GBS. However, being a case-patient was significantly associated with evidence of recent *Flavivirus* infection when combined with clinical criteria for suspected ZIKV disease (rash with at least two other ZIKV-like symptoms). At the time of assessment in that study, unlike the current study, all living GBS case-patients were at least five months out from neurologic symptom onset and the laboratory criteria were based on a recent *Flavivirus* infection (a positive or equivocal IgM test result for ZIKV or dengue). [13].

The seminal French-Polynesian study found a strong association between ZIKV and GBS [7], although similar to other assessments, the authors did not compare the strength of the GBS association with symptomatic ZIKV infection compared to asymptomatic ZIKV infection. Unlike the French Polynesia, Puerto Rico, and New Caledonia studies [7, 14, 15], our study does not support an association between ZIKV and GBS in Northeastern Mexico when using laboratory evidence of infection alone. However, other studies from Latin American and Asia Pacific do not show a significant association between ZIKV and GBS [16–18]. Methods and designs of these studies are heterogeneous, with differences in inclusion criteria and laboratory assays.

Other observational studies have also suggested a close association between GBS and ZIKV. In one Dutch study of cases returning from Suriname with ZIKV infection, one out of 18

patients (5.5%) developed GBS [19]. Also, in a cross-sectional study of 42 GBS cases in a region of Colombia, 40% had positive PCR and 32% had a positive anti-ZIKV IgM [20]. And yet, other similar reports have yielded contrasting results. For example, a recent report from Thailand, a country endemic for ZIKV, reported 1,417 cases of ZIKV infection but only two (0.14%) cases with concomitant GBS [21], a rate considerably lower than those previously observed in Polynesia and the Americas. In one early study of a ZIKV-infection outbreak in Yap (Micronesia), it was estimated that 73% of the population over three years of age had been infected (in a population of around 10,000 people), and no cases of GBS were reported [22]. Lastly, in a recent study carried out in the Gulf Mexican state of Veracruz, Mexico, where 28 cases of GBS were described, only two (7.1%) had positive anti-ZIKV IgG and none had positive IgM or PCR in serum [23]. And, although there are numerous case control studies related to Zika virus infections and GBS, our study is unusual and particularly valuable because it highlights differences between symptomatic vs asymptomatic Zika virus infections as they relate to GBS, not just the relationship of Zika virus infection in general to GBS.

Observational studies on the association of ZIKV and GBS have many limitations, and selection bias due to non-random selection is a significant issue, as is the loss of follow up and the lack of adjustment for overall ZIKV prevalence in a given region [24]. In this study, we did find a higher prevalence of positive ZIKV PCR compared to studies done in Puerto Rico [14], and French Polynesia [7].

Systematic reviews and meta-analyses of studies of ZIKV and GBS have been published. In a recent meta-analysis, from a total pooled number of 164,651 ZIKV-infected individuals, 1,513 developed ZIKV-associated GBS, 1.23% (95% CI = 1.17–1.29%) [25]. Another mathematical inference framework study utilizing data from 11 locations that had reported suspect ZIKV and GBS cases (including nine in the Americas), estimated that 2 (95% CI = 0.5–4.5) of reported GBS cases may occur per 10,000 ZIKV-infections [26].

ZIKV may be associated with particular phenotypic presentations of GBS. ZIKV-associated GBS has been associated with more dysautonomia, facial nerve palsy, and a more rapid onset of clinical GBS signs [27, 28]. Although our numbers are small, we found that dyspnea was more common in symptomatic ZIKV GBS case-patients, and symptomatic ZIKV infection was associated with more frequent need for mechanical ventilation.

This study is subject to several limitations. Using reports of antecedent illness may lead to several sources of bias, such as the non-specific nature of the symptoms, possible underreporting of acute illnesses by controls, and recall bias in reporting of symptoms by case-patients. Although selection of controls with non-febrile illness risks bias toward over-estimation of the significance of the predictive value of Zika-associated symptoms, analyses including only GBS case-patients who had no febrile illness within 5 days prior to onset of GBS were statistically significantly different from the controls. The limited number of subjects who had ZIKV-specific IgM antibodies tested for is another limitation, as is the inability to do serologic testing for DENV and CHIKV. The finding of up to 22% of GBS case-patients and controls having PCR-positivity for ZIKV was admittedly surprising; ordinarily, it would be expected that persons developing GBS would be outside of the time window for continuing to have ZIKV viremia. In the absence of confirmatory ZIKV-specific neutralization assay testing, we cannot say for certain that a certain amount of false-positivity may not have been present in our PCR results. However, the PCR positivity seemed specific for ZIKV; one might expect that if the problem was general false-positivity, one would observe unusually high percentages of DENV and CHIKV positivity as well, which was not the case. In addition, one might expect that false positivity would have been present in both GBS case-patients and controls; rather, the PCR results seemed preferentially present in the GBS case-patients rather than both case-patients and controls. Controls were obtained to account for geographic location, sex, and age, but

other factors such as socioeconomic condition were not controlled for and may have affected some of the findings. Finally, given a finding of 44% of case-patients reporting a diarrheal illness, we were unable to test for enteric pathogens, such as *Campylobacter jejuni*, which may have contributed to the overall burden of GBS in this group. The lack of a commercially available and standardized ELISA test for detecting anti-*Campylobacter* antibodies made pursuing this diagnosis logistically challenging.

## Conclusions

The accumulated evidence suggests a link between ZIKV infection and/or illness and GBS. Our study found a statistically significant association with symptomatic ZIKV but not with asymptomatic ZIKV infection alone. This finding supports a conclusion that the ZIKV association with GBS is stronger with ZIKV illness. Although the incidence of asymptomatic ZIKV is known to be several-fold higher than symptomatic ZIKV, our study suggests that during ZIKV epidemics, increases in GBS will occur primarily among those with antecedent symptomatic ZIKV.

## Acknowledgments

The authors would like to acknowledge the valuable contributions of the members of the Guillain-Barré syndrome -University Hospital -Mexican Institute of Social Security Working Group (GBS HU-IMSS WORKING GROUP): Alan Israel Benítez Álvarez, MD. Department of Neurology, University Hospital José Eleuterio González, Universidad Autonoma de Nuevo León, Monterrey, Nuevo León, Mexico; Alejandro Marfil-Rivera, MD. Department of Neurology, University Hospital José Eleuterio González, Universidad Autonoma de Nuevo León, Monterrey, Nuevo León, Mexico; Ana Sepúlveda-Núñez, MD. Epidemiology Division, UMAE Hospital de Especialidades No. 25, Instituto Mexicano del Seguro Social, Monterrey, Nuevo León, Mexico; Atzimba Álvarez Cortés, MD. Epidemiology Division, UMAE Hospital de Especialidades No. 25, Instituto Mexicano del Seguro Social, Monterrey, Nuevo León, Mexico; Beatriz Eugenia Chávez-Luevanos, MD. Department of Neurology, University Hospital José Eleuterio González, Universidad Autonoma de Nuevo León, Monterrey, Nuevo León, Mexico; Elsa Sierra-González, MD. Hospital General de Zona No. 33, Instituto Mexicano del Seguro Social, Monterrey, Nuevo León, Mexico; Gabriela López-Becerril, RN. Medical Research Unit for Neurological Diseases, UMAE Hospital de Especialidades, Centro Médico Nacional Siglo XXI, Instituto Mexicano del Seguro Social, Mexico City, Mexico; Gloria Nallely Velázquez-Castaño, MD. Department of Neurology, University Hospital José Eleuterio González, Universidad Autonoma de Nuevo León, Monterrey, Nuevo León, Mexico; Héctor Jorge Villarreal-Montemayor, MD. Department of Neurology, University Hospital José Eleuterio González, Universidad Autonoma de Nuevo León, Monterrey, Nuevo León, Mexico; Mallela Azuara-Castillo, MD. Hospital General de Zona No. 33, Instituto Mexicano del Seguro Social, Monterrey, Nuevo León, Mexico; María Garza-Sagástegui, MD. Medical Provisions Directorate of Nuevo León State. Instituto Mexicano del Seguro Social, Monterrey, Nuevo León, Mexico; Omar Israel Campos-Villarreal, MD. Department of Neurology, UMAE Hospital de Especialidades No. 25, Instituto Mexicano del Seguro Social, Monterrey, Nuevo León, Mexico; Roberto Corrales-Pérez, MD. Medical Epidemiology Assistance Coordinator of Nuevo León State, Instituto Mexicano del Seguro Social, Monterrey, Nuevo León, Mexico; Shaira España-Pérez, MD. Department of Neurology, University Hospital José Eleuterio González, Universidad Autonoma de Nuevo León, Monterrey, Nuevo León, Mexico; Víctor Hugo Borja-Aburto, MD, PhD. Primary Health-Care Unit. Instituto Mexicano del Seguro Social, Mexico City, Mexico.

## Author Contributions

**Conceptualization:** Fernando Gongora-Rivera, Israel Grijalva, Adrian Infante-Valenzuela, Carlos Cámara-Lemarroy, Jesus Soares, Joseph Y. Abrams, Ashley R. Styczynski, Ermias D. Belay, James J. Sejvar.

**Data curation:** Fernando Gongora-Rivera, Adrian Infante-Valenzuela, Carlos Cámara-Lemarroy, Elvira Garza-González, Martin Paredes-Cruz, Concepción Grajales-Muñiz, José Guerrero-Cantera, Ignacio Vargas-Ramos, Jesus Soares, Joseph Y. Abrams, James J. Sejvar.

**Formal analysis:** Fernando Gongora-Rivera, Adrian Infante-Valenzuela, Carlos Cámara-Lemarroy, Elvira Garza-González, Ignacio Vargas-Ramos, Joseph Y. Abrams.

**Funding acquisition:** Fernando Gongora-Rivera, Israel Grijalva, Margarita E. Villarino, Ermias D. Belay, James J. Sejvar.

**Investigation:** Fernando Gongora-Rivera, Israel Grijalva, Adrian Infante-Valenzuela, Elvira Garza-González, Martin Paredes-Cruz, Concepción Grajales-Muñiz, José Guerrero-Cantera, Ignacio Vargas-Ramos, Adrián Camacho-Ortiz, Margarita E. Villarino, James J. Sejvar.

**Methodology:** Fernando Gongora-Rivera, Israel Grijalva, Adrian Infante-Valenzuela, Carlos Cámara-Lemarroy, Jesus Soares, Ashley R. Styczynski, James J. Sejvar.

**Project administration:** Fernando Gongora-Rivera, Israel Grijalva, Adrian Infante-Valenzuela, Jesus Soares, James J. Sejvar.

**Resources:** Fernando Gongora-Rivera, Israel Grijalva, Adrian Infante-Valenzuela, Elvira Garza-González, Martin Paredes-Cruz, Ignacio Vargas-Ramos, Adrián Camacho-Ortiz, Margarita E. Villarino, James J. Sejvar.

**Supervision:** Fernando Gongora-Rivera, Israel Grijalva, Adrian Infante-Valenzuela, Elvira Garza-González, Jesus Soares, James J. Sejvar.

**Validation:** Fernando Gongora-Rivera, Israel Grijalva, Adrian Infante-Valenzuela, Elvira Garza-González, Ignacio Vargas-Ramos, Jesus Soares, Joseph Y. Abrams, Ashley R. Styczynski, James J. Sejvar.

**Writing – original draft:** Fernando Gongora-Rivera, Israel Grijalva, Carlos Cámara-Lemarroy, Elvira Garza-González, Jesus Soares, Joseph Y. Abrams, Lawrence B. Schonberger, James J. Sejvar.

**Writing – review & editing:** Fernando Gongora-Rivera, Israel Grijalva, Carlos Cámara-Lemarroy, Jesus Soares, Joseph Y. Abrams, Ashley R. Styczynski, Lawrence B. Schonberger, James J. Sejvar.

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
