## [Decision Letter · Decision Letter 0]

16 Dec 2019

PONE-D-19-30389

Zika Virus infection and Guillain-Barré syndrome in Northeastern Mexico: a case-control study

PLOS ONE

Dear Dr Sejvar,

Thank you very much for submitting your manuscript "Zika Virus infection and Guillain-Barré syndrome in Northeastern Mexico: a case-control study" (#PONE-D-19-30389) for review by PLOS ONE. As with all papers submitted to the journal, your manuscript was fully evaluated by academic editor (myself) and by independent peer reviewers. The reviewers appreciated the attention to an important health topic, but they raised substantial concerns about the paper that must be addressed before this manuscript can be accurately assessed for meeting the PLOS ONE criteria. Therefore, if you feel these issues can be adequately addressed, we invite you to submit a revised version of the manuscript that addresses the points raised during the review process. We can’t, of course, promise publication at that time.

We would appreciate receiving your revised manuscript by Dec 28 2019 11:59PM. To enhance the reproducibility of your results, we recommend that if applicable you deposit your laboratory protocols in protocols.io, where a protocol can be assigned its own identifier (DOI) such that it can be cited independently in the future. For instructions see: http://journals.plos.org/plosone/s/submission-guidelines#loc-laboratory-protocols

We look forward to receiving your revised manuscript.

Kind regards,

Abdallah M. Samy, PhD

Academic Editor

PLOS ONE

**Additional Editor Comments:**

I invited and received three reviews of your manuscript. All reviews raised substantial concerns about your manuscript as it currently stands. I read the manuscript myself and i must say that i completely agree that the reviews are detailed and solid. So, my decision is "major revision". Please address all these concerns before submitting a revised version of your manuscript. I would kindly ask the authors to confirm that they follow the Journal guidelines available via the link https://journals.plos.org/plosone/s/submission-guidelines. I would love to see the revised version of your manuscript as soon (i.e. within the next two weeks), so, we will save much time in the review process. Thanks too much for choosing PLOS ONE for your submission!

Journal Requirements:

4. One of the noted authors is a group or consortium: Guillain-Barré syndrome -University Hospital -Mexican Institute of Social Security Working Group (GBS HU-IMSS WORKING GROUP).

In addition to naming the author group, please list the individual authors and affiliations within this group in the acknowledgments section of your manuscript. Please also indicate clearly a lead author for this group along with a contact email address.

5. Your ethics statement must appear in the Methods section of your manuscript. If your ethics statement is written in any section besides the Methods, please move it to the Methods section and delete it from any other section. Please also ensure that your ethics statement is included in your manuscript, as the ethics section of your online submission will not be published alongside your manuscript.

**Reviewers' comments:**

Reviewer's Responses to Questions

**Comments to the Author**

1. Is the manuscript technically sound, and do the data support the conclusions?

Reviewer #1: Partly

Reviewer #2: Partly

Reviewer #3: Partly

2. Has the statistical analysis been performed appropriately and rigorously? 

Reviewer #1: Yes

Reviewer #2: No

Reviewer #3: Yes

3. Have the authors made all data underlying the findings in their manuscript fully available?

Reviewer #1: Yes

Reviewer #2: No

Reviewer #3: Yes

4. Is the manuscript presented in an intelligible fashion and written in standard English?

Reviewer #1: Yes

Reviewer #2: Yes

Reviewer #3: Yes

5. Review Comments to the Author

Reviewer #1: In this article, the authors included patients from three states of northeastern Mexico between August 2017 and June 2018, in order to establish whether the endemic circulation of Zika virus (ZIKV) was associated with the development of Guillaín-Barre syndrome (GBS). It is a case-control study, the cases corresponded to hospitalized or outpatient with GBS, with diagnostic certainty of grades 1 to 3 according to the classification of Brighton Collaboration Criteria. For each patient they included 3 controls (2.8) seen in emergency visits or hospitalized, with non-febrile illness for 48 hours, paired with cases by sex and age + minus 10 years, and seen in the same hospital within seven days of each case. After the interview, serum and urine were taken in both groups to determine exposure to ZIKV, dengue virus and Chikungunya virus by viral RNA amplification (PCR) and detection of IgM antibodies to ZIKV.

They studied 50 cases and 141 controls with similar demography. They found 22% of ZIKV positive PCR samples, both in cases and controls. The detection of IgM to ZIKV was performed in 26/50 (52%) of the cases and in 113/141 (80%) of the controls, with no difference in times between the onset of symptoms and the collection of samples.

The analysis prior to the neurological picture showed that the cases had with more frequency: Fever, diarrhea, myalgia, arthralgia, rash, conjunctivitis, headache, odynophagia and a composite outcome of typical ZIKV symptom (rash, joint pain, or conjunctivitis). The comparisons were carried out with conditional maximum likelihood estimation.

The first problem we encountered with the study is that the characteristics of the control group are not well described, at least in general terms. Such as the proportion of outpatients and hospitalized patients. It would be important to describe if in the control group there were patients with neurological disease, especially with GBS and, if among the emergency included patients, there were patients with minor trauma or trivial conditions and otherwise previously healthy, who would be representative of the “open” population. With some of these data, the general characteristics of the control group would be clearer.

Finally, with ZIKV, something similar to other viral diseases seems to happen in their post-epidemic transition to endemic status, and it is the fact that a high number of infections are asymptomatic as the clear example of the study of the island of Yap, where it was estimated that the 73% of the population was infected with ZIKV but only 18% of those infected (95% CI, 10 to 27), had a clinical illness that was probably attributable to ZIKV infection or 1 symptomatic person in 4.4 ZIKV infected persons (1).

A surprising finding in this study is that the controls had exactly the same percentage of positive PCR tests (22%) as the cases, which leads us to conclude on the importance of studying GBS patients as thoroughly as available resources allow it, to investigate coinfections, particularly gastrointestinal and thus be able to establish with greater solidity the association of ZIKV with GBS and the prevalence of other potential pathogens in endemic areas of ZIKV.

We recommend that a specific header must be opened prior to the conclusions, to clearly indicate the limitations of the study:

That electrophysiological studies were carried out in 48% of patients with GBS, that 20% of controls did not have the IgM antibody tests against ZIKV. That in the cases group, diarrhea occurred in 44% of the patients and it is not mentioned if the presence of intestinal pathogenic microorganisms was investigated, especially Campylobacter jejuni, which seems to be associated with Guillain Barré syndrome in the state of Veracruz, Mexico according to a publication by del Carpio-Orantes (2), this is a very important matter, which opens the possibility that the GBS was caused by concomitant enteric pathogens as an epiphenomenon to asymptomatic ZIKA virus infection as it would seem to occur in the controls.

Finally, there are some inaccuracies in the tables that merit a careful review:

1. In Table 3, in the line 9, conjunctivitis, the number of patients is 6, but the percentage in brackets is indicated 1 and should correspond to 12.

2. In Table 5. Neurological signs and symptoms at onset or nadir of GBS case-patients: n (%)

The first line reads neurological signs and symptoms all n = 50, the thrid column indicates Zika diagnosis by PCR or IgM (n = 26), and the fourth column shows Zika diagnosis by PCR or IgM and rash, joint pain, or conjunctivitis (n = 26), the sum is 52, no 50. In addition, if the individual data of the next line 10 + 16 +7 + 19 are added, it is equal to 52, so it should be reviewed why the total number of patients is exceeded by 2.

References

1. Duffy MR, Chen TH, Hancock WT, Powers AM, Kool JL, Lanciotti RS, Pretrick M,

Marfel M, Holzbauer S, Dubray C, Guillaumot L, Griggs A, Bel M, Lambert AJ, Laven

J, Kosoy O, Panella A, Biggerstaff BJ, Fischer M, Hayes EB. Zika virus outbreak

on Yap Island, Federated States of Micronesia. N Engl J Med. 2009 Jun

11;360(24):2536-43. doi: 10.1056/NEJMoa0805715. PubMed PMID: 19516034.

2. Del Carpio-Orantes L, Da Silva IRF, Moguel KGP, Díaz JSS, Del Pilar Mata

Miranda M, García-Méndez S, Perfecto-Arroyo MA, Solís-Sánchez I, Del Rosario

Pola-Ramírez M. Guillain Barré syndrome in arbovirus outbreak, Campylobacter claims

his throne. J Neurol Sci. 2019 Jan 15;396:254-255. doi: 10.1016/j.jns.2018.10.029.

Reviewer #2: Soares et al. describe a case-control study in Mexico assessing the association between Zika infection and GBS. They enroll 50 cases and match 1:3 141 controls to these. They assess Zika infection using RT-PCR in all and IgM in some patients. They find a similar and high proportion of Zika infection in both groups based on RT-PCR. Lab results in combination with at least one symptom does provide a signal that favors the hypothesis that there is an association between Zika infection and GBS.

From surveillance reports (PAHO, https://www.paho.org/hq/dmdocuments/2017/2017-phe-zika-situation-report-mex.pdf), it seems that the ZIKV outbreak took place in 2016, 2017. Could the authors in the introduction give some indication of the level of circulation, especially during the study period (August 2017, June 2018). Is this the tail of the epidemic, especially into 2018 was there any ZIKV circulation? How were the cases distributed over time? If the authors could provide an epidemic curve combined with when cases were sampled?

Regarding the diagnostic methods: The authors present a surprising proportion of RT-PCR positive patients both in the cases as control group (22%). We tend to believe that RT-PCR in general is highly specific, and that a positive result could be considered as Zika. False negatives are much more common, since, in settings as these, we are often too late. We know from GBS cases that were preceded by symptomatic ZIKV infection, that these would often occur 5-10 days after symptom onset. If we allow time before sampling these patients (here:5-52 days, we expect few to still have viral RNA in their blood or urine. Even in a peak of an epidemic, sampling symptomatic patients would likely not yield such high counts. Cross-validation of the results with a different method, IgM/neutralization would be of great value here to interpret the results. Can the authors clarify the diagnostics by at least providing crosstabulation with the IgM. It would be crucial to get more clarity on this issue, since this is one of the most relevant exposure assessments in the manuscript.

More specific: Should we indeed trust these results, or can the authors provide some additional verification of the results? (confirmation of the analyses, re-analyses?)

Much of the ‘significance’ relies on the combination of Lab results and one symptom (Rash, joint pain, conjunctivitis). Could the authors be more precise in explaining what these symptoms mean and how they were obtained: from clinical examination or the survey? E.g. does ‘joint pain’ mean that the subject had one day of joint pain 55 days before GBS onset (since the interview period mentions 2 months). A survey example could be provided in the supplementary material. Are these self-reported? What would classify as ‘conjunctivitis’ and what as ‘rash’? Do the authors agree that this information is crucial to interpret the likelihood of these (often aspecific) symptoms to be truly indicators for Zika virus infection.

How sensitive are the results to the selection of symptoms? Do combinations of symptoms, “at least two symptoms” still yield the same results?

Table 1: Could the number of IgM tested individuals and positive samples be added here? Crosstabulation of IgM and PCR would be of value as discussed above.

Table 2: the number of patients with IgM OR PCR (10) is lower than PCR only (11), that seems strange, since the first group would include at least the second.

Table 5: The sample size seems to be reduced to 26, although all patients have been tested using at least PCR? It seems that these are patients that have been assessed using IgM AND PCR instead IgM or PCR? What is the rationale to only take this subset here, where the other tables use the full sample as denominator? The conclusions seem to be thin, and based on multiple testing and wide confidence intervals. The text reports a ‘significant’ difference in dyspnea and a ‘trend’ in facial diplegia, dysphagia and dysarthria. What is a ‘trend’ and does the confidence interval take into account multiple testing? Would the authors consider phrasing these findings a bit more careful keeping in mind that these could as well be chance findings? E.g., we might as well say that there is a trend that PCR or IgM positivity (regardless of symptoms) is protective of GBS based on Table 4.

Specific comments:

Line 242-243: prevalence of RT-PCR positivity similar to Cao-Lormeau? They found 0/42 cases, this is not similar?

Line 248: that should be 2, not 2%?

Line 253: the ‘trend’ is here described as ‘slighlty more common’, what does this mean?

Line 262: The discussion of the RT-PCR results comes at a peculiar place in the discussion. This seems vital as this warrants care for the interpretation. It would be great to be clear and avoid the double negatives in sentence 266. You seem to state here that: “We are unsure about the validity of out RT-PCR results”.

It would be elegant to report according to a checklist like STROBE https://www.strobe-statement.org/index.php?id=available-checklists and provide the checklist as supplementary material.

Reviewer #3: The paper “Zika Virus infection and Guillain Barré syndrome in Northeastern Mexico: a case-control study”, authored by Gongora-Rivera F. et al., is an interesting and well written manuscript that assesses the relationship between the occurrence of Guillain Barré syndrome (GBS) and ZIKA virus infection. The authors did not find evidence of a link between laboratory evidence of ZIKA virus infection and the occurrence of GBS, but they did find a significant association when considering the antecedent of ZIKA’s typical signs and symptoms. I found the conclusion for this work supported previous evidence on this topic, assessing the impact on neurological complications related to acute infection with ZIKA virus in a specific population, in Northeast Mexico.

Nevertheless, I suggest some revisions to consider it for publication, as follows:

1-LINE 55: where it said, “We identified suspected GBS case-patients based on onset of compatible neurologic symptoms…” I suggest listing signs and symptoms considered to the case definition (can be in parentheses).

2-All sections of the manuscript where authors refer to the evaluation of exposure or serology for Dengue and Chikungunya (LINE 72: “…to determine exposure to ZIKV, DENV and CHIKV.”; LINE 94: “serological evidence of DENV, CHIKV and ZIKV infections.” need to be clarified and adjusted to the methodology applied to this work. Serological evidence of Dengue and Chikungunya virus was not evaluated. The authors tested for current infection with Zika, Dengue, and Chikungunya virus by RT-PCR, as well as to exposure to Zika with IgM antibodies against ZIKV.

3-LINE 93: List GBS risk factors (can be in parentheses).

4-LINE 107: Measures used to assess ZIKV status seem duplicative and create some confusion with respect to how the results are interpreted. For example, PCR-positive cases will be present also in the second group (Positive PCR assay or positive IgM assay) so that it is not clear how many patients were PCR- and IgM-positive, how many were only PCR-positive; and how many were only IgM-positive. Please clarify. This is very important since PCR and IgM positivity are not synonymous from a pathophysiological perspective, but rather reflect two different processes that can occur during the infection. It is my understanding that these criteria were chosen because not all cases and controls have a serology for ZIKV performed. This is unfortunate since GBS is considered a post-infectious complication such that antibodies rather than viremia seem more likely a measure of a post-infectious state. Still, this test was obtained in only half of the cases and 80% of controls. It will be very interesting to analyze the subgroup of patients with a serology test and its relation with GBS over the total of cases and controls with this test available, and also those patients with coexistence of PCR and IgM (if any). Therefore, authors can considerate these measures to assess ZIKV status: positive PCR; positive IgM; positive PCR and IgM.

5-Prolonged viremia is increasingly reported in ZIKV infection and had been related to pathogenesis [The Journal of Experimental Medicine (2019) 216 (10): 2302–2315]. Even if there is no way to know the duration of viremia in the cases with PCR positive presented in this work, do authors assume that the antecedent symptoms in the previous two months suggests prolonged viremia in cases with positive PCR?

RESULTS

6-Table 1. Results from the serology are missing. Please, add results from IgM assay against ZIKA. In a footnote it can be clarified for how many cases and how many controls serological testing was available. It also would be interesting express how many patients (if any?) were both PCR- and IgM- positive. Even when flavivirus infection is traditionally related to short viremias, followed by a rise of antibodies, some ZIKV infections have shown unusually prolonged viremias (more studies in pregnant woman). And even some recent studies link this prolonged viremia overlapping with peaking of specific antibodies, with the pathogenesis of congenital disease.

7-LINE 125 where it said “Recent infection by arbovirus…” must said “Current infection by arbovirus”. Since PCR refer to current infection, and IgM would refer to recent infection.

8-Table 2. I found this table very confusing to read, and I believe that confusion merge from the measures used to assess ZIKV status. I suggest follow directions previous suggest for METHODS.

9-Table 3. Fever, diarrhea and cough also had a correlation with cases. What do the authors think about that? Where these symptoms in relation with typical Zika symptoms, or they were observed in different patients? (and diarrhea is misspelled in the table)

10-Table 4. It is not clear. Again, this dual measure of ZIKV status causes confusion. Same comment to table 2.

11-Table 5. The same comment to previous tables. I suggest one block: Zika diagnosis by PCR and or IgM (n50) with two sub columns: Zika positive and Zika negative. Each one of these with two sub columns: with and without typical zika symptoms. Include (N and %) at the headline.

12-Table 6. Same comment table 5.

CONCLUSIONS

13-The authors develop a complete and correct analysis of appropriate bibliography. I consider the main limitation of this work, the lack of serology test for ZIKV for almost half of the cases (and actually, it is hard to find in the manuscript, how many of cases and controls had a positive result). This limitation is briefly mention in the discussion (LINE 261-262). Because GBS is a post-infectious event, is expected to find this clinical manifestation in synchrony with the presence of antibodies. It would be interesting to evaluate if exist an association with the presence of IgM against ZIKV and GBS, considering only that subgroup of cases and controls.

14-Another important limitation is the lack of Dengue and Chikungunya serology. The co-circulation of these Aedes borne diseases in the region, and the almost impossibility to clinical differentiate the clinical features make so relevant these tests. This limitation needs to be mention, and clarity it in methods as it was mentioned before.

15-I found very interesting the evaluation of previous typical ZIKV symptoms mainly for its moderate Positive Predictive Value for GBS diagnosis. The PPV could be calculated.

6. PLOS authors have the option to publish the peer review history of their article (what does this mean?). If published, this will include your full peer review and any attached files.

Reviewer #1: Yes: José Luis Soto-Hernandez M.D.

Reviewer #2: No

Reviewer #3: No

---

## [Author Response · Author response to Decision Letter 0]

28 Jan 2020

Get Outlook for iOS

From: em.pone.0.67f9a8.15841351@editorialmanager.com <em.pone.0.67f9a8.15841351@editorialmanager.com> on behalf of PLOS ONE <em@editorialmanager.com>

Sent: Monday, December 16, 2019 12:10:37 PM

To: Sejvar, James (CDC/DDID/NCEZID/DHCPP) <zea3@cdc.gov>

Subject: PLOS ONE Decision: Revision required [PONE-D-19-30389] - [EMID:e47842d417be5818] 

PONE-D-19-30389

Zika Virus infection and Guillain-Barré syndrome in Northeastern Mexico: a case-control study

PLOS ONE

Dear Dr Sejvar,

Thank you very much for submitting your manuscript "Zika Virus infection and Guillain-Barré syndrome in Northeastern Mexico: a case-control study" (#PONE-D-19-30389) for review by PLOS ONE. As with all papers submitted to the journal, your manuscript was fully evaluated by academic editor (myself) and by independent peer reviewers. The reviewers appreciated the attention to an important health topic, but they raised substantial concerns about the paper that must be addressed before this manuscript can be accurately assessed for meeting the PLOS ONE criteria. Therefore, if you feel these issues can be adequately addressed, we invite you to submit a revised version of the manuscript that addresses the points raised during the review process. We can’t, of course, promise publication at that time.

We would appreciate receiving your revised manuscript by Dec 28 2019 11:59PM. To enhance the reproducibility of your results, we recommend that if applicable you deposit your laboratory protocols in protocols.io, where a protocol can be assigned its own identifier (DOI) such that it can be cited independently in the future. For instructions see: http://journals.plos.org/plosone/s/submission-guidelines#loc-laboratory-protocols

• A rebuttal letter that responds to each point raised by the academic editor and reviewer(s). This letter should be uploaded as separate file and labeled 'Response to Reviewers'.

• A marked-up copy of your manuscript that highlights changes made to the original version. This file should be uploaded as separate file and labeled 'Revised Manuscript with Track Changes'.

• An unmarked version of your revised paper without tracked changes. This file should be uploaded as separate file and labeled 'Manuscript'.

We look forward to receiving your revised manuscript.

Kind regards,

Abdallah M. Samy, PhD

Academic Editor

PLOS ONE

Additional Editor Comments:

I invited and received three reviews of your manuscript. All reviews raised substantial concerns about your manuscript as it currently stands. I read the manuscript myself and i must say that i completely agree that the reviews are detailed and solid. So, my decision is "major revision". Please address all these concerns before submitting a revised version of your manuscript. I would kindly ask the authors to confirm that they follow the Journal guidelines available via the link https://journals.plos.org/plosone/s/submission-guidelines. I would love to see the revised version of your manuscript as soon (i.e. within the next two weeks), so, we will save much time in the review process. Thanks too much for choosing PLOS ONE for your submission!

Journal Requirements:

 JJS: We have reviewed the style requirements, and believe that our manuscript does meet these requirements.

 JJS: We are happy to have our study data available; we have indicated the point of contact to access the complete, de-identified data.

 JJS: The corresponding author’s ORCID id is valid. 0000-0002-2536-3276.

4. One of the noted authors is a group or consortium: Guillain-Barré syndrome -University Hospital -Mexican Institute of Social Security Working Group (GBS HU-IMSS WORKING GROUP).

In addition to naming the author group, please list the individual authors and affiliations within this group in the acknowledgments section of your manuscript. Please also indicate clearly a lead author for this group along with a contact email address.

 JJS: We have named a lead author for the GBS HU-IMSS Working Group. It is the following, along with email address: 

GBS HU IMSS Working Group

Hector Jorge Villarreal, MD

Chief of Servicio de Neurología del Hospital Universitario, Monterrey. Mexico 

Email: neurologia01@yahoo.com.mx

5. Your ethics statement must appear in the Methods section of your manuscript. If your ethics statement is written in any section besides the Methods, please move it to the Methods section and delete it from any other section. Please also ensure that your ethics statement is included in your manuscript, as the ethics section of your online submission will not be published alongside your manuscript.

 JJS: This has been done

Reviewers' comments:

Reviewer's Responses to Questions

Comments to the Author

1. Is the manuscript technically sound, and do the data support the conclusions?

Reviewer #1: Partly

Reviewer #2: Partly

Reviewer #3: Partly

2. Has the statistical analysis been performed appropriately and rigorously? 

Reviewer #1: Yes

Reviewer #2: No

Reviewer #3: Yes

3. Have the authors made all data underlying the findings in their manuscript fully available?

Reviewer #1: Yes

Reviewer #2: No

Reviewer #3: Yes

4. Is the manuscript presented in an intelligible fashion and written in standard English?

Reviewer #1: Yes

Reviewer #2: Yes

Reviewer #3: Yes

5. Review Comments to the Author

Reviewer #1: In this article, the authors included patients from three states of northeastern Mexico between August 2017 and June 2018, in order to establish whether the endemic circulation of Zika virus (ZIKV) was associated with the development of Guillaín-Barre syndrome (GBS). It is a case-control study, the cases corresponded to hospitalized or outpatient with GBS, with diagnostic certainty of grades 1 to 3 according to the classification of Brighton Collaboration Criteria. For each patient they included 3 controls (2.8) seen in emergency visits or hospitalized, with non-febrile illness for 48 hours, paired with cases by sex and age + minus 10 years, and seen in the same hospital within seven days of each case. After the interview, serum and urine were taken in both groups to determine exposure to ZIKV, dengue virus and Chikungunya virus by viral RNA amplification (PCR) and detection of IgM antibodies to ZIKV.

They studied 50 cases and 141 controls with similar demography. They found 22% of ZIKV positive PCR samples, both in cases and controls. The detection of IgM to ZIKV was performed in 26/50 (52%) of the cases and in 113/141 (80%) of the controls, with no difference in times between the onset of symptoms and the collection of samples.

The analysis prior to the neurological picture showed that the cases had with more frequency: Fever, diarrhea, myalgia, arthralgia, rash, conjunctivitis, headache, odynophagia and a composite outcome of typical ZIKV symptom (rash, joint pain, or conjunctivitis). The comparisons were carried out with conditional maximum likelihood estimation.

The first problem we encountered with the study is that the characteristics of the control group are not well described, at least in general terms. Such as the proportion of outpatients and hospitalized patients. It would be important to describe if in the control group there were patients with neurological disease, especially with GBS and, if among the emergency included patients, there were patients with minor trauma or trivial conditions and otherwise previously healthy, who would be representative of the “open” population. With some of these data, the general characteristics of the control group would be clearer.

 JJS: We appreciate the comment by the reviewer. Of the 141 controls, 24 were outpatients, and 117 were hospitalized. This would suggest that the control population would not be an accurate reflection of the nonhospitalized population. We have included this information in the results.

Finally, with ZIKV, something similar to other viral diseases seems to happen in their post-epidemic transition to endemic status, and it is the fact that a high number of infections are asymptomatic as the clear example of the study of the island of Yap, where it was estimated that the 73% of the population was infected with ZIKV but only 18% of those infected (95% CI, 10 to 27), had a clinical illness that was probably attributable to ZIKV infection or 1 symptomatic person in 4.4 ZIKV infected persons (1).

JJS: We concur with the reviewer on this.

A surprising finding in this study is that the controls had exactly the same percentage of positive PCR tests (22%) as the cases, which leads us to conclude on the importance of studying GBS patients as thoroughly as available resources allow it, to investigate coinfections, particularly gastrointestinal and thus be able to establish with greater solidity the association of ZIKV with GBS and the prevalence of other potential pathogens in endemic areas of ZIKV.

JJS: The authors concur with the reviewer that other, alternative etiologies of GBS with a temporal association, such as Campylobacter jejuni, ideally would ideally have been tested for an excluded. However, the lack of a widely available and standardized ELISA serum assay for anti-Campylobacter antibodies made such a diagnosis challenging in this situation.

We recommend that a specific header must be opened prior to the conclusions, to clearly indicate the limitations of the study: 

That electrophysiological studies were carried out in 48% of patients with GBS, that 20% of controls did not have the IgM antibody tests against ZIKV. That in the cases group, diarrhea occurred in 44% of the patients and it is not mentioned if the presence of intestinal pathogenic microorganisms was investigated, especially Campylobacter jejuni, which seems to be associated with Guillain Barré syndrome in the state of Veracruz, Mexico according to a publication by del Carpio-Orantes (2), this is a very important matter, which opens the possibility that the GBS was caused by concomitant enteric pathogens as an epiphenomenon to asymptomatic ZIKA virus infection as it would seem to occur in the controls.

JJS: We have included a section on limitations and will ensure that language to this effect will be placed in the limitations: ‘Diarrhea was present in approximately 44% of cases. Diarrhea is a common clinical manifestation of the gastrointestinal bacterium Campylobacter jejuni, which also has a strong association with the axonal form of GBS, acute motor axonal neuropathy (AMAN). However, the lack of a commercially available and standardized ELISA test for detecting anti-Campylobacter antibodies made pursuing this diagnosis logistically challenging. ‘ 

Finally, there are some inaccuracies in the tables that merit a careful review:

1. In Table 3, in the line 9, conjunctivitis, the number of patients is 6, but the percentage in brackets is indicated 1 and should correspond to 12.

 JJS: This has been fixed in the manuscript.

2. In Table 5. Neurological signs and symptoms at onset or nadir of GBS case-patients: n (%)

The first line reads neurological signs and symptoms all n = 50, the thrid column indicates Zika diagnosis by PCR or IgM (n = 26), and the fourth column shows Zika diagnosis by PCR or IgM and rash, joint pain, or conjunctivitis (n = 26), the sum is 52, no 50. In addition, if the individual data of the next line 10 + 16 +7 + 19 are added, it is equal to 52, so it should be reviewed why the total number of patients is exceeded by 2.

 JJS: We added a footnote to clarify this. Specifically, this represents 26 people of the 50 who had IgM testing done; the two columns represent separate groups of 26. Thus, the sum of the two columns should not add up to 52. 

References

1. Duffy MR, Chen TH, Hancock WT, Powers AM, Kool JL, Lanciotti RS, Pretrick M,

Marfel M, Holzbauer S, Dubray C, Guillaumot L, Griggs A, Bel M, Lambert AJ, Laven

J, Kosoy O, Panella A, Biggerstaff BJ, Fischer M, Hayes EB. Zika virus outbreak

on Yap Island, Federated States of Micronesia. N Engl J Med. 2009 Jun

11;360(24):2536-43. doi: 10.1056/NEJMoa0805715. PubMed PMID: 19516034.

2. Del Carpio-Orantes L, Da Silva IRF, Moguel KGP, Díaz JSS, Del Pilar Mata

Miranda M, García-Méndez S, Perfecto-Arroyo MA, Solís-Sánchez I, Del Rosario

Pola-Ramírez M. Guillain Barré syndrome in arbovirus outbreak, Campylobacter claims

his throne. J Neurol Sci. 2019 Jan 15;396:254-255. doi: 10.1016/j.jns.2018.10.029.

Reviewer #2: Soares et al. describe a case-control study in Mexico assessing the association between Zika infection and GBS. They enroll 50 cases and match 1:3 141 controls to these. They assess Zika infection using RT-PCR in all and IgM in some patients. They find a similar and high proportion of Zika infection in both groups based on RT-PCR. Lab results in combination with at least one symptom does provide a signal that favors the hypothesis that there is an association between Zika infection and GBS.

From surveillance reports (PAHO, https://www.paho.org/hq/dmdocuments/2017/2017-phe-zika-situation-report-mex.pdf), it seems that the ZIKV outbreak took place in 2016, 2017. Could the authors in the introduction give some indication of the level of circulation, especially during the study period (August 2017, June 2018). Is this the tail of the epidemic, especially into 2018 was there any ZIKV circulation? How were the cases distributed over time? If the authors could provide an epidemic curve combined with when cases were sampled?

JJS: As the reviewer rightly points out, part of the investigation was conducted during periods of relatively low, and even absent, Zika transmission. While we would like to give the readers a sense of the burden of Zika virus infection over the study period, unfortunately accurate, complete, and consistent data do not exist for these time periods in the Monterrey region, or for Mexico in general. Based upon the data that we were able to glean, Zika was still circulating during 2016 -2018, and based on what we know from 2016-2018, it seems like Zika incidence was highest during the fall through early winter, which seems to correspond roughly to an increase in GBS cases (and subsequent controls). We appreciate the reviewer’s request for an epidemiologic curve, but when we formatted one, it did not appear to very well address the question posed by the reviewer, and we chose to omit it. 

Regarding the diagnostic methods: The authors present a surprising proportion of RT-PCR positive patients both in the cases as control group (22%). We tend to believe that RT-PCR in general is highly specific, and that a positive result could be considered as Zika. False negatives are much more common, since, in settings as these, we are often too late. We know from GBS cases that were preceded by symptomatic ZIKV infection, that these would often occur 5-10 days after symptom onset. If we allow time before sampling these patients (here:5-52 days, we expect few to still have viral RNA in their blood or urine. Even in a peak of an epidemic, sampling symptomatic patients would likely not yield such high counts. Cross-validation of the results with a different method, IgM/neutralization would be of great value here to interpret the results. Can the authors clarify the diagnostics by at least providing crosstabulation with the IgM. It would be crucial to get more clarity on this issue, since this is one of the most relevant exposure assessments in the manuscript.

JJS: We concur with the reviewer that the proportion / percentage of cases and controls having positive results by PCR testing was much higher than expected. We also agree that it is more likely to have false negatives by PCR due to the timing issue. Unfortunately, during this investigation, there was so little sample left after routine bloodwork and PCR testing that we were, in the vast majority of cases, unable to ‘confirm’ the PCR results with concomitant serology. However, the definitive testing to substantiate this is not feasible.

More specific: Should we indeed trust these results, or can the authors provide some additional verification of the results? (confirmation of the analyses, re-analyses?)

JJS: Again, unfortunately we are unable to provide additional verification of results, due to lack of sample. The reason that the controls were just as likely to have a positive PCR result is unclear.

Much of the ‘significance’ relies on the combination of Lab results and one symptom (Rash, joint pain, conjunctivitis). Could the authors be more precise in explaining what these symptoms mean and how they were obtained: from clinical examination or the survey? E.g. does ‘joint pain’ mean that the subject had one day of joint pain 55 days before GBS onset (since the interview period mentions 2 months). A survey example could be provided in the supplementary material. Are these self-reported? What would classify as ‘conjunctivitis’ and what as ‘rash’? Do the authors agree that this information is crucial to interpret the likelihood of these (often aspecific) symptoms to be truly indicators for Zika virus infection.

How sensitive are the results to the selection of symptoms? Do combinations of symptoms, “at least two symptoms” still yield the same results?

JJS: We thank the reviewer for this observation. Indeed, these were self-reported signs and symptoms reported among subjects. Per request, a copy of the case report form has been added to the supplementary material. ‘Conjunctivitis’ and ‘Rash’ were whatever was interpreted by the subject, but it is generally thought that most lay persons recognize a rash and conjunctivitis (red eyes). 

When combining symptoms, we unfortunately found that there was an insufficient number of patients in each cell with 2 or more symptoms to evaluate this. Essentially, there were 1 of 26 cases, and 0 of 113 controls that are Zika positive and have 2 or more symptoms. This gives an uninterpretable odds ratio with an interval of 0 to infinity. 

Table 1: Could the number of IgM tested individuals and positive samples be added here? Crosstabulation of IgM and PCR would be of value as discussed above.

JJS: Please see added New Table 2

Table 2: the number of patients with IgM OR PCR (10) is lower than PCR only (11), that seems strange, since the first group would include at least the second.

JJS: This is due to the fact that only 26 of the 50 cases received an IgM test. We restricted the assessment of Zika positivity by PCR or IgM to those 26 patients (this was to remove the possibility of bias from including all 50 patients: those who received an IgM test would be inherently more likely to be reported as Zika+, and if they were different in any way from the 24 who did not receive an IgM test, this could result in unforeseen biases).

Table 5: The sample size seems to be reduced to 26, although all patients have been tested using at least PCR? It seems that these are patients that have been assessed using IgM AND PCR instead IgM or PCR? What is the rationale to only take this subset here, where the other tables use the full sample as denominator? The conclusions seem to be thin, and based on multiple testing and wide confidence intervals. The text reports a ‘significant’ difference in dyspnea and a ‘trend’ in facial diplegia, dysphagia and dysarthria. What is a ‘trend’ and does the confidence interval take into account multiple testing? Would the authors consider phrasing these findings a bit more careful keeping in mind that these could as well be chance findings? E.g., we might as well say that there is a trend that PCR or IgM positivity (regardless of symptoms) is protective of GBS based on Table 4.

JJS: We added a footnote to clarify this. We have removed the term ‘trend’ and excluded facial diplegia, dysphagia, and dysarthria, mentioning only dyspnea that is ‘significant’ statistically, though we acknowledge the wide confidence intervals.

Specific comments:

Line 242-243: prevalence of RT-PCR positivity similar to Cao-Lormeau? They found 0/42 cases, this is not similar?

JJS: This was clearly an error on our part, and this has been corrected in the text.

Line 248: that should be 2, not 2%?

JJS: The reviewer is correct; this number should be 2, not 2%. This has been corrected.

Line 253: the ‘trend’ is here described as ‘slighlty more common’, what does this mean?

JJS: This has been corrected to indicate that only dyspnea was associated with a statistically significant measure of being more common among cases than controls (page 15 line 243, clean version).

Line 262: The discussion of the RT-PCR results comes at a peculiar place in the discussion. This seems vital as this warrants care for the interpretation. It would be great to be clear and avoid the double negatives in sentence 266. You seem to state here that: “We are unsure about the validity of out RT-PCR results”.

JJS: We apologize to the reviewer, but we cannot locate the sentence in which there is a double negative; we are simply saying that we cannot determine with absolute certainty that there were not some subjects with false-positive results. But then we lay out why we feel that the results are sound.

It would be elegant to report according to a checklist like STROBE https://www.strobe-statement.org/index.php?id=available-checklists and provide the checklist as supplementary material.

JJS: Per the reviewer’s suggestion, we have utilized the STROBE checklist for this paper. We feel that criteria 1 – 7, 9-12, and 14-22 of the STROBE checklist for case-control studies was adequately addressed by our paper. 

Reviewer #3: The paper “Zika Virus infection and Guillain Barré syndrome in Northeastern Mexico: a case-control study”, authored by Gongora-Rivera F. et al., is an interesting and well written manuscript that assesses the relationship between the occurrence of Guillain Barré syndrome (GBS) and ZIKA virus infection. The authors did not find evidence of a link between laboratory evidence of ZIKA virus infection and the occurrence of GBS, but they did find a significant association when considering the antecedent of ZIKA’s typical signs and symptoms. I found the conclusion for this work supported previous evidence on this topic, assessing the impact on neurological complications related to acute infection with ZIKA virus in a specific population, in Northeast Mexico.

Nevertheless, I suggest some revisions to consider it for publication, as follows:

1-LINE 55: where it said, “We identified suspected GBS case-patients based on onset of compatible neurologic symptoms…” I suggest listing signs and symptoms considered to the case definition (can be in parentheses).

JJS: Per the reviewer’s request, we have added some signs and symptoms compatible with GBS (page 5, line 104).

2-All sections of the manuscript where authors refer to the evaluation of exposure or serology for Dengue and Chikungunya (LINE 72: “…to determine exposure to ZIKV, DENV and CHIKV.”; LINE 94: “serological evidence of DENV, CHIKV and ZIKV infections.” need to be clarified and adjusted to the methodology applied to this work. Serological evidence of Dengue and Chikungunya virus was not evaluated. The authors tested for current infection with Zika, Dengue, and Chikungunya virus by RT-PCR, as well as to exposure to Zika with IgM antibodies against ZIKV.

JJS: We concur with the reviewer that this investigation did not assess for serologic evidence of DENV and CHIKV, and that this was misstated in the manuscript. This has been corrected. 

3-LINE 93: List GBS risk factors (can be in parentheses).

JJS: This has been added (page 6, line 117)

4-LINE 107: Measures used to assess ZIKV status seem duplicative and create some confusion with respect to how the results are interpreted. For example, PCR-positive cases will be present also in the second group (Positive PCR assay or positive IgM assay) so that it is not clear how many patients were PCR- and IgM-positive, how many were only PCR-positive; and how many were only IgM-positive. Please clarify.

JJS: We concur with the reviewer that this information is important to the interpretation of the results here. We have defined, in the table, the number of subjects that were PCR-positive only; IgM-positive only; and those that were PCR- and IgM-positive. (See New Table 2)

 This is very important since PCR and IgM positivity are not synonymous from a pathophysiological perspective, but rather reflect two different processes that can occur during the infection. It is my understanding that these criteria were chosen because not all cases and controls have a serology for ZIKV performed. This is unfortunate since GBS is considered a post-infectious complication such that antibodies rather than viremia seem more likely a measure of a post-infectious state. Still, this test was obtained in only half of the cases and 80% of controls. It will be very interesting to analyze the subgroup of patients with a serology test and its relation with GBS over the total of cases and controls with this test available, and also those patients with coexistence of PCR and IgM (if any). Therefore, authors can considerate these measures to assess ZIKV status: positive PCR; positive IgM; positive PCR and IgM.

JJS: We concur with the reviewer that PCR and IgM represent very different pathophysiological underpinnings of flavivirus infection, and that, as a post-infectious phenomenon, GBS would more likely be expected to be positive by IgM, rather than PCR, testing. So, per above, we are including the breakdown of PCR+, IgM+, and PCR / IgM+. This can be found in New Table 2.

5-Prolonged viremia is increasingly reported in ZIKV infection and had been related to pathogenesis [The Journal of Experimental Medicine (2019) 216 (10): 2302–2315]. Even if there is no way to know the duration of viremia in the cases with PCR positive presented in this work, do authors assume that the antecedent symptoms in the previous two months suggests prolonged viremia in cases with positive PCR?

JJS: We are aware of the reports of prolonged viremia, although the reports in the literature are relatively few. The reviewer is correct that we are unable to say with any degree of certainty the duration of viremia in our cases. We cannot, however, a priori assume that this is reflective of a prolonged viremia. 

RESULTS

6-Table 1. Results from the serology are missing. Please, add results from IgM assay against ZIKA. In a footnote it can be clarified for how many cases and how many controls serological testing was available. It also would be interesting express how many patients (if any?) were both PCR- and IgM- positive. Even when flavivirus infection is traditionally related to short viremias, followed by a rise of antibodies, some ZIKV infections have shown unusually prolonged viremias (more studies in pregnant woman). And even some recent studies link this prolonged viremia overlapping with peaking of specific antibodies, with the pathogenesis of congenital disease.

JJS: The IgM results have been added to table 1, along with footnotes that specify the number of cases (26) and controls (113) that received IgM tests

7-LINE 125 where it said “Recent infection by arbovirus…” must said “Current infection by arbovirus”. Since PCR refer to current infection, and IgM would refer to recent infection.

JJS: This has been corrected. 

8-Table 2. I found this table very confusing to read, and I believe that confusion merge from the measures used to assess ZIKV status. I suggest follow directions previous suggest for METHODS.

JJS: We edited Table 2 (now Table 3) so that the number of cases assessed for each group is easier to understand

9-Table 3. Fever, diarrhea and cough also had a correlation with cases. What do the authors think about that? Where these symptoms in relation with typical Zika symptoms, or they were observed in different patients? (and diarrhea is misspelled in the table)

JJS: Fever, diarrhea, and cough are very nonspecific findings of acute viral infection. Although it could be argued that rash, joint pain, and conjunctivitis are similarly nonspecific, during our several investigations into the clinical characteristics of Zika virus infection, it seemed that these three symptoms were strikingly more common in Zika virus infection as opposed to other general viral infections. Thus, we placed more emphasis on these three symptoms of ZIKV.

‘Diarrhea’ has been corrected.

10-Table 4. It is not clear. Again, this dual measure of ZIKV status causes confusion. Same comment to table 2.

JJS: We edited Table 4 (now Table 5) so that the number of cases assessed for each group is easier to understand; specifically, we added headings of “All Observations” and “Patients Undergoing IgM Test”.

11-Table 5. The same comment to previous tables. I suggest one block: Zika diagnosis by PCR and or IgM (n50) with two sub columns: Zika positive and Zika negative. Each one of these with two sub columns: with and without typical zika symptoms. Include (N and %) at the headline.

JJS: The main purpose of Table 5 (now Table 6) is to determine if patients with Zika-associated GBS were clinically different from those with non-Zika-associated GBS. This was done using two slightly different definitions for Zika diagnosis (Zika diagnosis by PCR or IgM, or Zika diagnosis by PCR or IgM and rash, joint pain, or conjunctivitis). The suggested changes would alter the comparisons: we would be seeing if, out of patients testing positive for Zika by PCR or IgM, whether those who had previous Zika symptoms differed clinically from those who did not have previous Zika symptoms (this also asks of those who tested negative for Zika by PCR or IgM, whether those who had previous Zika symptoms differed clinically from those who did not have previous Zika symptoms). These were not the research questions we were interested in. For the sake of clarity, we did add a footnote that specified which patients were included in these analyses. 

12-Table 6. Same comment table 5.

JJS: Similar to Table 5, changing Table 6 (now Table 7) in the suggested manner would have changed the research question. We did add the same footnote from Table 5 in Table 6 for clarification.

CONCLUSIONS

13-The authors develop a complete and correct analysis of appropriate bibliography. I consider the main limitation of this work, the lack of serology test for ZIKV for almost half of the cases (and actually, it is hard to find in the manuscript, how many of cases and controls had a positive result). This limitation is briefly mention in the discussion (LINE 261-262). Because GBS is a post-infectious event, is expected to find this clinical manifestation in synchrony with the presence of antibodies. It would be interesting to evaluate if exist an association with the presence of IgM against ZIKV and GBS, considering only that subgroup of cases and controls.

JJS: Unfortunately, there are very few patients who are Zika+ by IgM. Of the cases, 3 of 26 (11.5%) had a positive IgM test; for controls, it was 9 of 113 (8.0%), for an odds ratio of 1.50 (0.37 – 5.95). Thus, we were unable to assess this question.

14-Another important limitation is the lack of Dengue and Chikungunya serology. The co-circulation of these Aedes borne diseases in the region, and the almost impossibility to clinical differentiate the clinical features make so relevant these tests. This limitation needs to be mention, and clarity it in methods as it was mentioned before.

JJS: We concur with the reviewer that the clinical signs and symptoms of dengue, Chikungunya, and Zika can be clinically indistinguishable. And, had we the necessary aliquots of serum to conduct these investigations, we would most certainly have done so. We were, however, limited by the availability of the necessary aliquots of serum. Nonetheless, there was no clear evidence of a great deal of circulating dengue or Chikungunya virus in Monterrey at the time, as evidenced by lack of PCR findings for both these viruses. We have, however, mentioned in the discussion / limitations that this does represent a substantial limitation in our findings (page 18, line 406). It has similarly been clarified in the methods. 

15-I found very interesting the evaluation of previous typical ZIKV symptoms mainly for its moderate Positive Predictive Value for GBS diagnosis. The PPV could be calculated.

 JJS: While a PPV can technically be calculated, this is a case-control study, with the number of cases and controls purposefully selected by the study investigators; a PPV calculated from this study has no relation to a PPV applicable to the general population. However, of 25 patients with one of the typical Zika symptoms, 16 were cases (PPV=64%). Of the 11 patients with a Zika symptom and Zika+ by PCR or IgM, 7 were cases (PPV = 64%).

---

## [Decision Letter · Decision Letter 1]

24 Feb 2020

Zika Virus infection and Guillain-Barré syndrome in Northeastern Mexico: a case-control study

PONE-D-19-30389R1

Dear Dr. Sejvar,

We are pleased to inform you that your manuscript has been judged scientifically suitable for publication and will be formally accepted for publication once it complies with all outstanding technical requirements.

With kind regards,

Abdallah M. Samy, PhD

Academic Editor

PLOS ONE

---

## [Editor Report · Acceptance letter]

12 Mar 2020

PONE-D-19-30389R1 

Zika Virus infection and Guillain-Barré syndrome in Northeastern Mexico: a case-control study 

Dear Dr. Sejvar:

I am pleased to inform you that your manuscript has been deemed suitable for publication in PLOS ONE. Congratulations! Your manuscript is now with our production department. 

With kind regards,

on behalf of

Dr. Abdallah M. Samy 

Academic Editor

PLOS ONE